# Physicochemical Changes Occurring during Long-Time Fermentation of the Indigenous Alcoholic Sorghum-Based Beverages Brewed in Northern Cameroon

**James Ronald Bayoï** [1,*] **and François-Xavier Etoa** [2]

1   Laboratory of Biochemistry and Microbiology, Department of Biological Sciences, University of Maroua, Maroua 814, Cameroon

2   Laboratory of Microbiology, Department of Microbiology, University of Yaoundé I, Yaoundé 812, Cameroon; fxetoa@yahoo.fr

*   Correspondence: jabar982002@gmail.com; Tel.: +237-699-458-665

**Abstract:** In Cameroon, alcoholic beverages represent one of the main consumed drinks. In northern regions, indigenous sorghum beers are very popular and widely consumed in an actively fermenting state by people. In this study, some physicochemical parameters of alcoholic sorghum beverages and correlations between them were evaluated during fermentation for 10 days. The indigenous white and red beers were produced at the laboratory scale assisted by experimental producers and some parameters (pH, total acidity, alcohol, sugars, density, total solids, temperature, and conductivity) were measured on the wort and fermented beverages. The pH decreased from 3.2 to 2.4 and 3.11 to 2.41; total acidity increased from 1.07 to 5.1 g/L and 0.5 to 4.6 g/L; alcohol was enhanced from 0 to 9.5% and 0 to 6.8% (*v/v*); total solids dropped from 13.6 to 5°P and 12.2 to 3.3°P, respectively, in the white and red sorghum beers. The multivariate analysis showed a good correlation between consumption of sugar, the decrease in total solids and density with the decrease in pH. Additionally, it was shown that a perfect link exists between the production of alcohol and organic acids. The hierarchical analysis showed that indigenous red beer samples fermented for one and two days and those fermented for four to 10 days were related and could be separate in two distinct groups, whereas white turbid beer samples were separated in three different groups, those fermented for one to four days, five to six days, and seven to 10 days. The results obtained could serve as a guide to better understand the fermentation process of indigenous alcoholic sorghum-based beverages.

**Keywords:** Cameroon; sudano-sahelian zone; sorghum; long-time fermentation; turbid beers; physicochemical changes; clustering

## 1. Introduction

Fermented beverages have a long tradition and contribute to the nutrition of many societies and cultures worldwide. Traditional fermentation was empirically developed in ancient times as a process of raw food preservation, and at the same time, the production of new foods with different sensorial characteristics and enhanced nutritional value [1]. Based on archaeological and archaeobotanical findings, it is generally believed that over 9000 years ago, individuals around the globe were already fermenting beverages [2]. It has been reported that fermented foods and beverages globally contribute 20 to 40% of the food supply and usually occupy the third position of foods consumed by man [3]. It is therefore not surprising that fermented foods and beverages make a big contribution to people's diets in Africa [4]. Cereals such as sorghum, pearl millet, and maize are generally used for the production of indigenous fermented beverages that are widely consumed all over the African continent [5]. It is estimated that over 60 million people living in the very hot, drought-prone tropical regions of Africa use sorghum and millet as part of their staple diet [6,7]. In Cameroon, sorghum is essentially cultivated in the soudano-sahelian zone with an annual average production of about 500,000 tons [8], which places

the northern part of the country at the top rank of cereal production. Due to the many problems of storage encountered in northern Cameroon, a significant part of the sorghum produced is used to brew indigenous alcoholic beverages known as *bil bil*, which accounts for approximately 80% of the total consumption of alcoholic beverages in these regions of the country [9]. These are popular because of the social, religious, nutritional, and therapeutic values that are associated with them [10]. They are cherished by both rural and urban populations because they are less costly and available everywhere throughout the year [10]. In the northern part of Cameroon, indigenous sorghum-based alcoholic beverages are sold as street food at some original and convivial places called cabarets or *saré* in the local dialect [11]. In the famous "Mandara" mountainous range, situated in the far north of Cameroon, two particular and highly culturally symbolic homebrewed sorghum beers are produced and known as red *tè* beer (or male beer) and white *mpedli* beer (or female beer) [12,13]. These beers are an integral part of the socio-cultural life of the *kapsiki*, an ethnic tribe belonging to the *Mandara* group. In addition to their colors, which are different, the two *kapsiki* beers are produced by two distinct artisanal fermentation processes. The white beer is made by fermenting the mixture obtained from brewed malt flour added to the cooked non-malted flour of sorghum (*fufu*) without the inoculation of any starter culture. The red beer is generally produced by fermenting sweetish wort (*tè kwarhèni*) from malted flour of sorghum using an artisanal starter culture. Previous studies on the microbiological and physicochemical characteristics of both indigenous beers as well as their artisanal processing technologies have been carried out and are well documented. Contrary to European beers made with barley and fermented by selected yeasts during a long time ranging between eight to 15 days, indigenous sorghum beers are fermented by artisanal leaven for a time that varies between 10 h to 48 h. As a result, these traditional sorghum beers from northern Cameroon are consumed in an actively fermenting state and the beverages contain large amounts of fragments of insoluble materials. Thus far, all data from both indigenous beers have provided an overview of the properties (or qualities) of alcoholic beverages only at the end of the fermentation step and at the moment of their consumption. However, physicochemical analysis of indigenous alcoholic sorghum-based beverages during a long fermentation time has not been documented.

Given that understanding of changes in the physicochemical characteristics is essential to upgrade the traditional processing to commercial scale, this paper aimed to highlight the physicochemical changes that occurred during 10 days of fermentation of the indigenous alcoholic beverages brewed in northern Cameroon and to determine the correlations among physicochemical parameters and the similarity between the different indigenous beers based on fermentation time using multivariate analysis. The importance of this study is to upgrade the production of the indigenous and culturally embedded beverages to the next level, which is a small industrial scale.

## 2. Materials and Methods

### 2.1. Vegetal Material and Artisanal Starter

The local sorghum (*Sorghum bicolor* L. Moench) grains of the red variety and the white variety were purchased from the local market of Maroua, Diamaré, Cameroon. The plant materials were transported to the Laboratory of Food Technology in a plastic bag and subsequently verified by the botanical experts of the Department of Biological Sciences of the Faculty of Science. The artisanal starter used for fermentation of the red kapsiki beer was a gifted dried powder graciously given to us by the local beer producers from Mogode, Mayo-Tsanaga, Cameroon.

### 2.2. Analysis of Quality of the Sorghum Grains

To ensure the production of the indigenous beers with good quality, we performed quality control of the sorghum grains used during processing. Physicochemical and technological parameters of sorghum grains such as water content, germination capacity, germination energy, the temperature of germination, percentage of impurity, and 1000-kernel

weight were evaluated using the method described by Analytica-EBC [14]. After analysis, only grains with good physicochemical and technological characteristics were used for artisanal processing of the traditional sorghum-based fermented alcoholic beverages.

### 2.3. Laboratory Preparation of the Indigenous Alcoholic Beverages

Two of the experimented women previously interviewed for the description of the artisanal processing of both indigenous beers were called to assist us during the production of the kapsiki beers according to the manufacturing flowchart described in our previous studies [12,13] and following their observations. The production was conducted in the Laboratory of Food Technology, Institute of Agricultural Research, Unit of Maroua, Cameroon.

### 2.4. Laboratory Processing of the Sorghum Turbid White Beer

The traditional processing involved stages of malting, brewing, and fermentation. Many (about a sixth of the total grains used for the beer production) of the white grains of sorghum (0.8 kg) was soaked in clean water (2.5 L) and left to settle for 24 h at room temperature (about 29 °C). The grains were germinated in the dark for two days and sprayed with water every 12 h, kilned for 10 h at 40 °C using a hot-air drier (Memmert GmbH + Co. KG, Büchenbach, Germany), ground to malt sorghum flour with an electrical grinder (Duronic, UK), and kept for further use. The greatest quantity of the remaining white sorghum grains (4.2 kg) was cleaned and mashed in non-malted sorghum flour. The flour collected was soaked in water (10 L) for 72 h and the floury pellet (6.4 kg) recovered was mixed with distilled clean water (12 L). The sorghum paste was cooked between 78 °C and 95 °C for about 3 h to obtain a cooked dough called *"fufu"*. After cooling, all the white sorghum malt flour kept was added and mixed to the roasted sorghum dough. The mixture was hand-kneaded until a wort was obtained and left for spontaneous fermentation [12].

### 2.5. Laboratory Processing of the Sorghum Turbid Red Beer

First, the red sorghum grains (5 kg) were soaked in clean water (15 L) at room temperature (28 °C–29 °C) for 24 h, and germinated in the dark for four days with a water spray every 12 h. The germinated red sorghum grains were kilned for 10 h at 40 °C using a hot-air drier (Memmert GmbH + Co. KG, Büchenbach, Germany). After milling of the dried grains with an electrical grinder (Duronic, UK), the malted flour obtained was mixed with distilled clean water (20 L). The mixture obtained was decanted for 3 h at 45 °C using a water bath (Memmert GmbH + Co. KG, Büchenbach, Germany), then the supernatant (11 L) was separated and kept at room temperature. The bottom was pre-cooked between 70 °C and 90 °C for 4 h while shaking every 5 min, later mixed with the previous supernatant to give a sour wort after a night spontaneous fermentation. The sour wort was boiled for 7 h at 103 °C and was mixed every 5 min to give sweet wort at the end. The artisanal dried starter was mixed with distilled water and the sweet-sour wort was inoculated with up to 10% (*v/v*) artisanal starter suspension for alcoholic fermentation [13].

### 2.6. Yield of Wort Production

The yield of wort production was determined by the ratio between the quantity of worth produced and the quantity of raw plant material used according to Equation (1) below:

$$Yp~(\%) = (Mn/Mr) \times 100 \qquad (1)$$

where Mn is the mass of wort obtained (g); Mr is the mass of raw material used (g); and Yp is the yield of wort production.

### 2.7. Samples Preparation and Fermentation Monitoring

One hundred (100) milliliters of sample for each indigenous beer produced previously were withdrawn eleven times in triplicate at 0, 1, 2, 3, 4, 5, 6, 7, 8, 9, and 10 days of fermentation for a total of 33 samples collected for each artisanal fermented alcoholic

beverage. Samples were immediately centrifuged for 5 min at $1000\times g$ and the supernatant was successively filtered through filter paper (Whatman #1) and a membrane filter (0.62 μm diameter). The beer filtrates collected were used for physicochemical analysis for the determination of pH, total acidity, Brix, sugar, density, alcohol content, conductivity, and temperature.

*2.8. Physicochemical Analyses of the Traditional Sorghum-Based Alcoholic Beverages*

The pH and temperature of the samples were recorded before centrifugation and immediately after harvesting using respectively an ATC portable pH-meter (Eco Testr, Singapore) and infrared thermometer (Initio Jeulin 251040, Montpellier, France). The total soluble solids was expressed as degree Brix, the density and conductivity of filtered supernatant were measured using portable devices, respectively an ATC refractometer (RHB 90, Shenzhen, China), an aerometer (Assistent 6105/3, Shenzhen, China), and a conductimeter (Eco Testr, Singapore). Total sugar of the filtrates was evaluated by applying the phenol-sulfuric colorimetric method as proposed by Dubois et al. [15]. The optical density values recorded at the 490 nm wavelength were measured by using a UV–Vis spectrophotometer (Jenway 7305, Bibby Scientific, Group HQ, Staffordshire, UK).

The alcohol content of the filtrates was determined with the combined results of degree Brix and density according to Equation (2), as described by Pauline et al. [16] with slight modifications:

$$\text{Alcohol content } (\%, v/v) = \text{DB} - [(\text{D} - 1) \times 1000] \tag{2}$$

where DB is the degree Brix value (°B) and D is the density.

Total acidity expressed as percentage (%) of acetic acid and malic acid in the filtrates was titrated against a 0.1 N NaOH solution using phenolphthalein (0.1% $w/v$ in ethanol) as the colored indicator. Equation (3) was used to determine the total acidity as described by Desobgo et al. [17]:

$$\text{Total acidity} = [(\text{normality of 0.1 NaOH} \times \text{mL volume used of 0.1 NaOH} \times 1000)/(\text{mL volume of filtrate supernatant})] \times \text{k} \tag{3}$$

where k is equal at 0.067 for malic acid and 0.8 for acetic acid.

*2.9. Statistical Analysis*

All measurements were done in triplicate and the results were presented as mean $\pm$ standard deviation using curves and bar charts to illustrate changes in the physicochemical parameters during fermentation. These parameters were subjected to one-way analysis of variance (ANOVA) to determine the mean differences among the beer samples fermented at different times. Whenever significant differences in ANOVA ($p < 0.05$) were detected, the HSD (Honest Significant Difference) Tukey's multiple range test [18] was applied to discriminate pairs of means significantly different at $p < 0.05$ in STAGRAPHICS software centurion version 16.1 (Technologies Inc., Virginia, USA). Multivariate analysis, especially principal component analysis (PCA) and hierarchical cluster analysis (HCA) [19], were performed to analyze the correlation between different physicochemical parameters of beer samples, and the relationship between both indigenous beer samples using the SPSS statistical program (SPSS20, IBM Inc., Armonk, New York, USA).

**3. Results**

*3.1. Physicochemical and Technological Characteristics of Sorghum Grains*

The characteristics of sorghum grains used for the processing of indigenous beers are summarized in Table 1. The impurity percent of red (2.5%) and white (2.4%) sorghum grains was not significantly ($p < 0.05$) different. The water content of grains varied from 5.84 to 5.92% and was lower than the standards recommended by brewers (13%). The weight of white grains was significantly higher than the weight of red grains. The weight of 1000

grains ranged between 33.5 g and 42.2 g. The grains of the red sorghum variety showed a germination capacity (97.8%) and germination energy (69.0%) significantly higher than those of white sorghum grains (85.8% and 55%). Both sorghum grains showed relatively the same temperature during the germination step.

**Table 1.** Physicochemical characteristics of some grains of sorghum sampled in the main markets of Maroua town and used for the production of the indigenous beers.

| Varieties | Impurity (%) | H (%) | GC (%) | $GE_4$ (%) | $GE_8$ (%) | $W_{1000}$ (g) | TG (°C) |
|---|---|---|---|---|---|---|---|
| White sorghum | 2.4 ± 0.1 [a] | 5.8 ± 0.1 [a] | 85.8 ± 1.1 [a] | 55.0 ± 0.0 [a] | 100.0 ± 0.0 [a] | 42.2 ± 1.1 [a] | 32.3±1.0 [a] |
| Red sorghum | 2.5 ± 0.2 [a] | 5.9 ± 0.2 [a] | 97.8 ± 1.1 [b] | 69.0 ± 1.4 [b] | 87.5 ± 3.5 [b] | 33.5 ± 0.5 [b] | 32.6 ±1.2 [a] |

H: water content; GC: germinative capacity of 200 grains; GE: germinative energy of 100 grains by using 4 mL of distilled water ($GE4$) or 8 mL of distilled water ($GE_8$); $W_{1000}$: weight of 1000 grains; TG: Temperature of grains during the step of germination. Mean values preceded by one common letter in the same column were not significantly different ($p < 5\%$).

### 3.2. Processing of Indigenous Beers

Table 2 presents the yield of wort production during the processing of red and white traditional beers. We observed that the wort yield of white indigenous beer (59%) was significantly lower than the wort yield of red opaque beer (62%).

**Table 2.** Wort yield according to the type of sorghum used for the production of indigenous alcoholic beverages.

| Indigenous Beer | White Beer | Red Beer |
|---|---|---|
| Yield of production (%) | 59.8 ± 2.9 [a] | 62.6 ± 3.8 [b] |

Mean values ($n = 3$ repetitions) preceded by one common letter (a, b) were not significantly different ($p < 5\%$).

### 3.3. Evolution of Physicochemical Parameters during Fermentation of Opaque Beers

Changes in pH during fermentation of the indigenous beers wort are illustrated in Figure 1. We observed that the initial pH values were 3.2 and 3.11, respectively, for white and red home-brewed beers. After the first day of fermentation, we had a significant pH drop in the white and red beers at 2.41 and 2.52, respectively. From day 2 to day 6, we noted no significant increase in pH values for both beers, followed by a second slight drop of pH values between day 7 and day 10 of fermentation for the red beer. The alcohol content of the white and red beers home-brewed in northern Cameroon is shown in Figure 2. Alcohol production during fermentation of white grains described a sigmoid curve with a minimal value of 0% and maximal value of 9.5% after 10 days of fermentation. We had a significant increase in the alcohol content from the first to the third day of fermentation (1.9–4%) and that remained steady between the third and fifth day of fermentation (4–3.7%). From the sixth to the eighth day, we noted a second and significant rise in alcohol content (4.9–8.5%) in the white beer samples with a steadiness until the tenth day. The alcohol content during fermentation of the red beer described a curve with two dip points. We noticed a significant enhancement in alcohol content after the first day of fermentation of the red opaque beer (2.8%), followed by a dip and relative constancy from the second to the fifth day of fermentation (3.3–4.2%). Then, we observed a second dip on the sixth day (4.7%), followed by a second phase of significant increase in alcohol content between the seventh and eighth day of fermentation (4.7–7.1%).

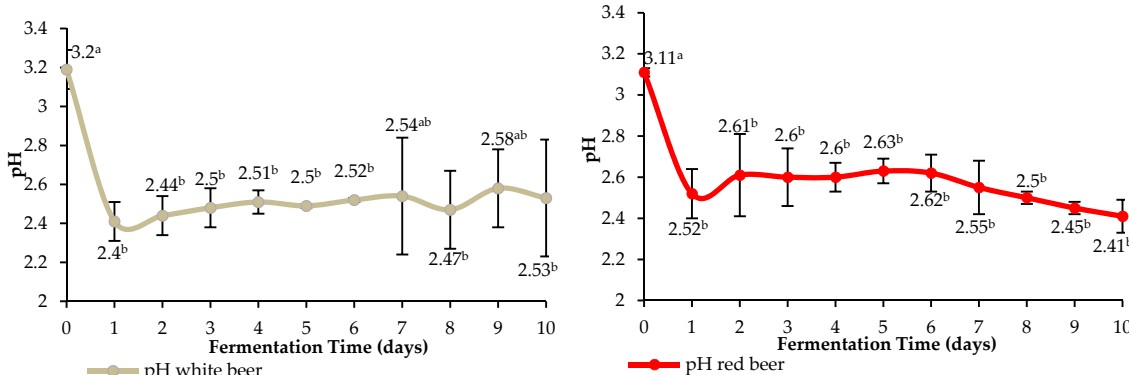

**Figure 1.** Evolution of pH during fermentation of the indigenous sorghum beers. Mean values (*n* = 3 repetitions) preceded by at least one common letter (a, b) were not significantly different (*p* < 0.05). Bars represent standard deviations.

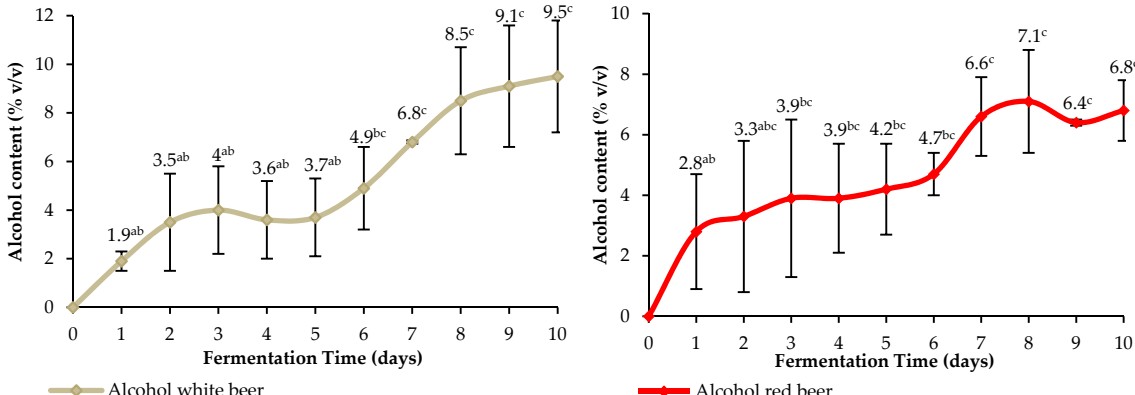

**Figure 2.** Evolution of alcohol content during fermentation of the indigenous sorghum beers. Mean values (*n* = 3 repetitions) preceded by at least one common letter (a, b, c) were not significantly different (*p* < 0.05). Bars represent standard deviations.

The total solids and sugar contents of indigenous beer samples decreased with time of fermentation (Figure 3). The total solids and sugars of the white beer samples varied from 13.6°P to 5°P and 143.3 g/L to 59 g/L, respectively. During fermentation, the red home-brewed beer samples showed a total solids content varying between 12.2°P to 3.3°P and total sugars ranged between 132 g/L to 44 g/L. The changes in density during the fermentation process of the home-brewed beers are presented in Figure 4. We noticed a significant decrease in density with time of fermentation for both indigenous beers. Density values of the white beer samples varied from 1.06 to 1.025 g/L and from 1.053 to 1.017 g/L for red beer samples, respectively, between days 0 to 10 of the fermentation process. We observed a constant density in the beer from the sixth day until the tenth day. Figure 5 shows changes in temperature during the fermentation of sorghum beers. Temperature of the white beer samples during the fermentation process oscillated between 31.4 °C before fermentation and 34.8 °C on the tenth day of fermentation. The same evolution was observed with the temperature of the red beer samples, which moved from 31.7 °C at the initial time to 33.6 °C at the end of the fermentation process. However, we observed a significant enhancement in temperature after the first day of fermentation (35.6 °C), following a sudden drop from the second day up to the seventh day. Titrable acetic and malic acid contents during fermentation of the opaque sorghum beers are shown in Figure 6. We observed a significant increase in acetic and malic acids after a day of fermentation of the white beer (50 g/L and 4.2 g/L) and red beer (43.3 g/L and 3.6 g/L) samples, followed by a constancy up to the tenth day of fermentation of the white beer (only for malic acid) and red beer (both organic acids). We noticed a second increase in acetic acid content of the white beer samples from the seventh day of fermentation (60.1 g/L).

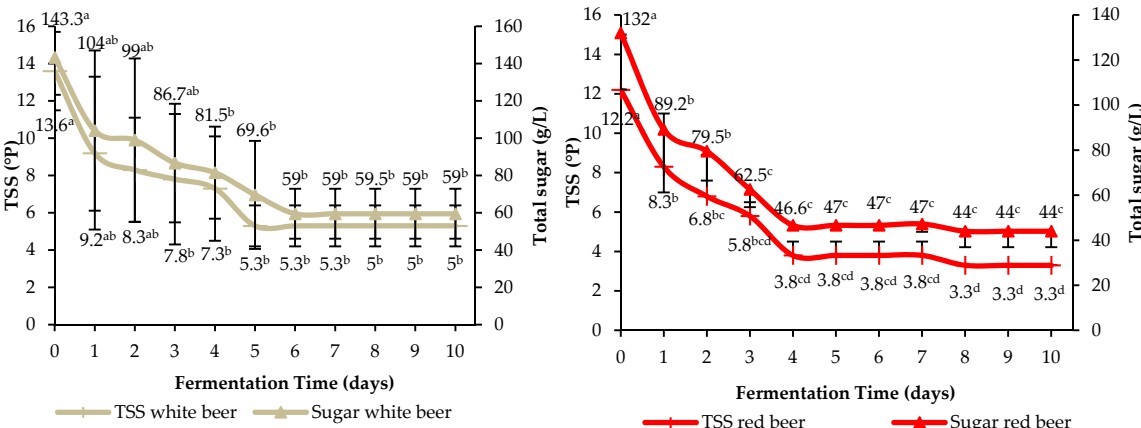

**Figure 3.** Evolution of total soluble solids and sugars during fermentation of the indigenous sorghum beers. Mean values (*n* = 3 repetitions) preceded by at least one common letter (a, b, c, d) were not significantly different (*p* < 0.05). Bars represent standard deviations.

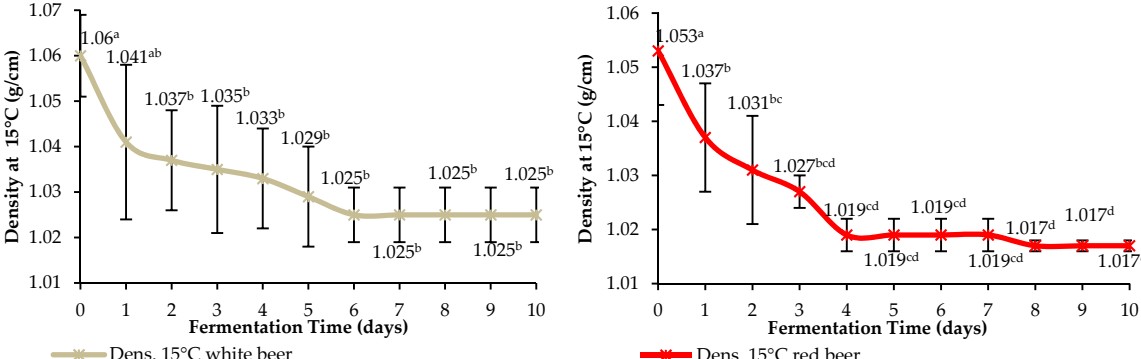

**Figure 4.** Evolution of density during fermentation of the indigenous sorghum beers. Mean values (*n* = 3 repetitions) preceded by at least one common letter (a, b, c, d) were not significantly different (*p* < 0.05). Bars represent standard deviations.

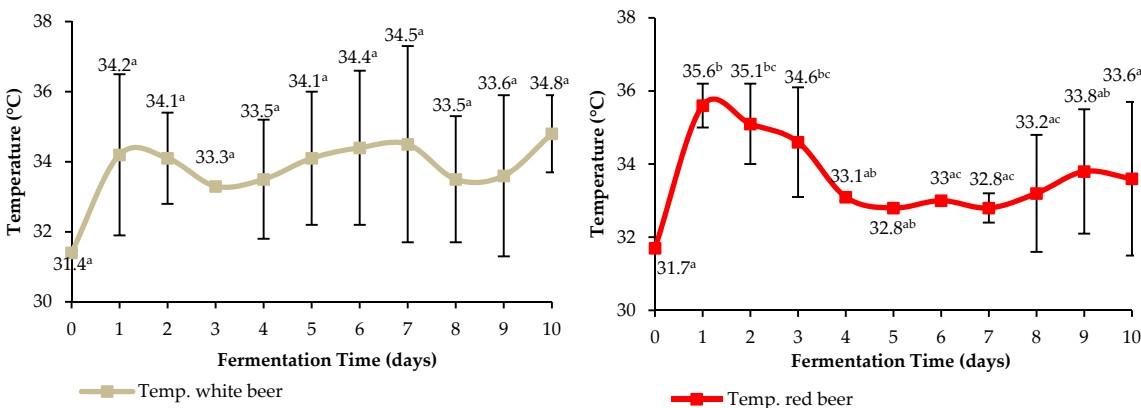

**Figure 5.** Evolution of temperature during fermentation of the indigenous sorghum beers. Mean values (*n* = 3 repetitions) preceded by at least one common letter (a, b, c) were not significantly different (*p* < 0.05). Bars represent standard deviations.

The conductivity of the white and red beer samples is illustrated in Figure 7. The statistical analysis showed that there was no significant difference between the conductivity values of both beer samples during the fermentation process. The conductivity values varied between 800 μS/cm and 920 μS/cm for the white beer samples and from 898 μS/cm to 1073 μS/cm for the red beer samples.

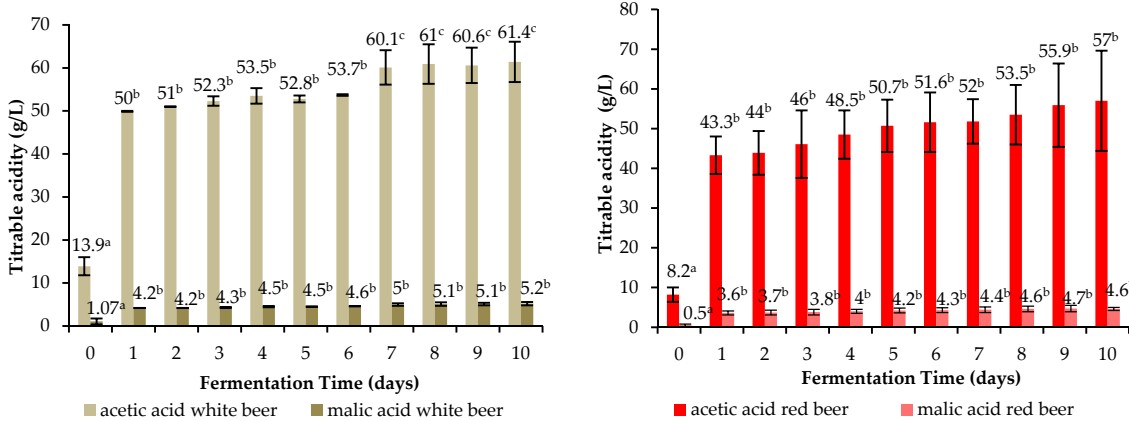

**Figure 6.** Evolution of titrable acidity during fermentation of the indigenous sorghum beers. Mean values (*n* = 3 repetitions) preceded by at least one common letter (a, b, c) were not significantly different (*p* < 0.05). Bars represent standard deviations.

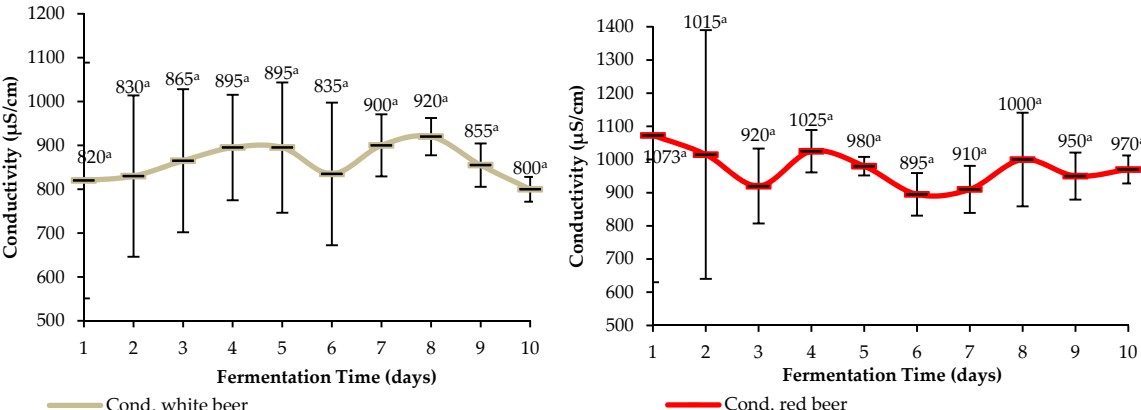

**Figure 7.** Evolution of conductivity during fermentation of the indigenous sorghum beers. Mean values (*n* = 3 repetitions) preceded by at least one common letter were not significantly different (*p* < 0.05). Bars represent standard deviations.

### 3.4. Multivariate Analysis of Indigenous Alcoholic Beverages during Fermentation

To facilitate the interpretation of the correlations between variables evaluated during fermentation of both traditional beers used, principal component analysis (PCA) was performed with nine physicochemical parameters of indigenous beer samples. As shown in Figure 8, Kaiser–Meyer–Olkin (KMO) measures (0.67–0.721) and Bartlett's test of sphericity (*p*-value = 0.000) confirmed that sampling was adequate and the principal component (or factor) analysis was valid. The nine physicochemical variables were reduced to two principal components (F1 and F2) that had eigenvalues larger than one and retained for rotation. F1 accounted for 75.7% and 76.7%, whereas F2 only accounted for 13.64% and 17% of the total variations, respectively, for the white and red indigenous beers. Combination of F1 and F2 together explained 88.34% and 93.75% of the total variance, respectively, for home-brewed white beer and red beer samples.

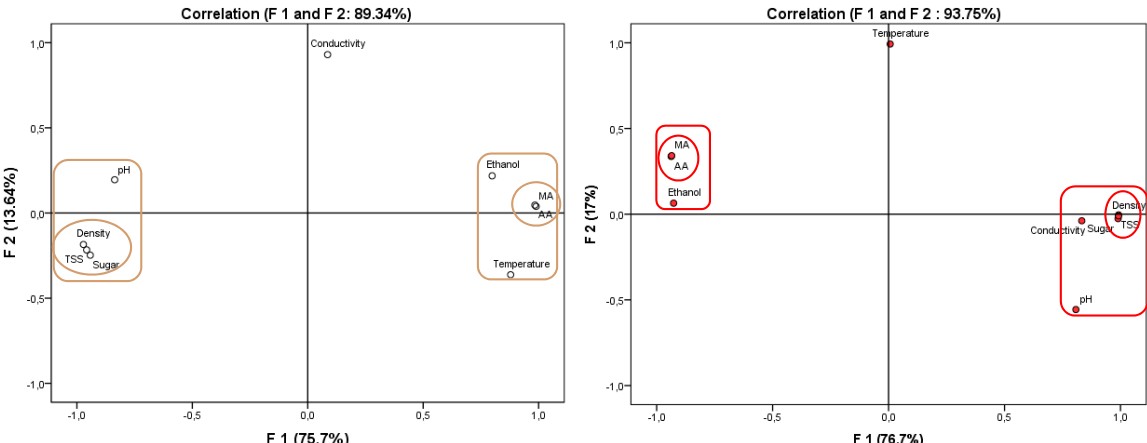

**Figure 8.** Correlation loadings plot of the two main factors resulting from principal component analysis of physicochemical parameters during fermentation of the white beer (**left**) and red beer (**right**) for ten days. Notes: Rectangles and circles show $R^2$ = 50% and 98%, respectively. Rotation method: Varimax with Kaiser normalization (KMO = 0.721 and 0.67, respectively, for white and red beer samples; *p*-value = 0.000). TSS: total soluble solids; AA: acetic acid; MA: malic acid.

The geometrical figures indicate the correlation levels between physicochemical variables. Rectangles indicate only 50% of the explained variance whereas circles indicate up to 98% of the variance explained linearly and showed a strong correlation between physicochemical regarded variables. The majority of the variation captured by F1 for all the physicochemical parameters served to distinguish the beer samples during fermentation. The PCA showed that with the traditional white beer, ethanol, temperature, and titrable acidity (expressed as acetic and malic acids) were positively loaded on F1 and sugar, TSS, density, and pH were negatively loaded on the same factor. Conductivity was the only variable positively loaded on F2. In contrast, the factor analysis of the indigenous red beer samples indicated that variables such as sugar, TSS, density, pH, and conductivity positively contributed to F1 and variables such as titrable acidity (expressed as acetic and malic acids) and ethanol were negatively loaded on F1. Only temperature was positively loaded on F2. Figure 9 presents the scores plots derived from the measured physicochemical variables of beer during fermentation and showed clear discrimination of the observations. However, the plot of the white beer samples appeared to be in contrast to the plot of red beer samples. For example, the white beer samples after one and two days of fermentation were on the negative side of F2, whereas the red beer samples at the same time of fermentation were found to be on the positive side of F2. This was always the case with beer samples fermented for six, seven, nine, and 10 days, which were found on the negative side and the positive side of F1 for indigenous white beer and red beer, respectively.

In an attempt to simplify the interpretation of relationships between indigenous beer samples during fermentation, hierarchical clustering analysis (HCA) was applied to the physicochemical variables by squared Euclidean measurement. The centered-reduced normalization was applied to improve the classification of the samples of both indigenous sorghum-based alcoholic beverages. As shown in Figure 10, two main and distinct clusters were identified: I and II. The beer samples were divided into seven sub-groups. The red beer samples recovered after one and two days of fermentation were classified into the same sub-cluster (E) while the red beer sampled after four to 10 days of fermentation belonged to a different sub-cluster (A). The white beer samples recovered between one to four days, and seven to 10 days of fermentation were grouped into two distinct sub-clusters of D and B, respectively. However, the white beer samples recovered after five and six days of fermentation and the red beer samples obtained after three days of fermentation were grouped into the same sub-cluster (C). Hierarchical analysis showed that though the indigenous beer samples belonged to the same cluster, the white beer and red beer before the fermentation process could be presented as two different sub-clusters (F and G).

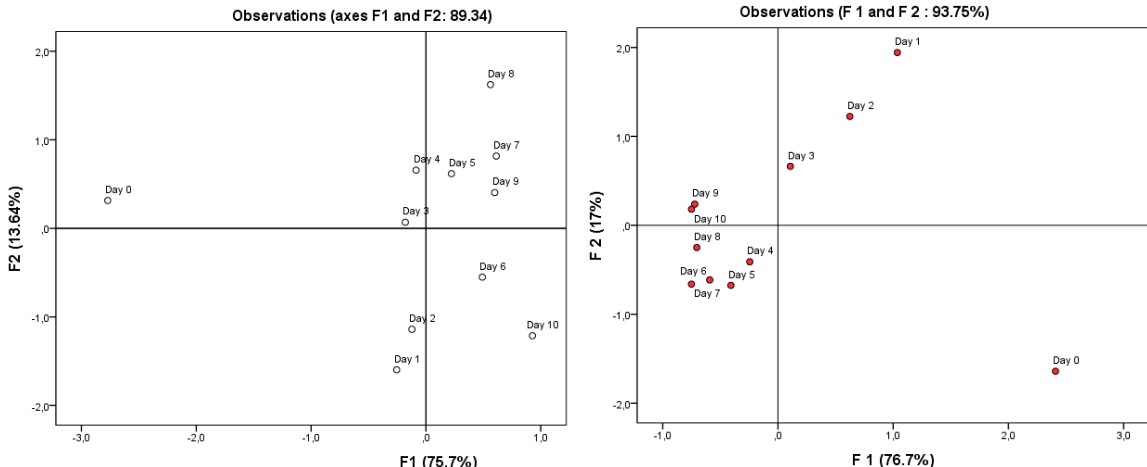

**Figure 9.** Principal Component Analysis scores plots derived using the physicochemical values of indigenous white beer (**left**) and red beer (**right**) samples for ten days of fermentation.

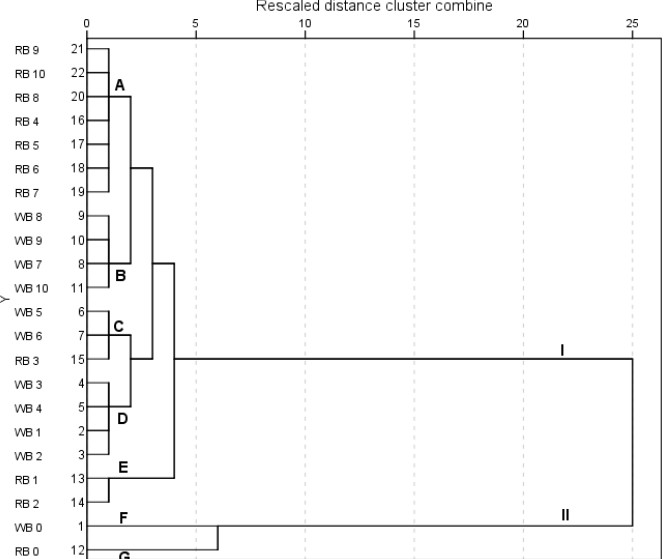

**Figure 10.** Clustering dendrogram of the indigenous sorghum beer samples at different fermentation times performed with hierarchical cluster analysis using the centroid method and squared Euclidean measurement. Notes: x-axis and y-axis mean the rescaled distance cluster and the beer samples measured, respectively. WB: white beer; RB: red beer. Arabic numerals (1 to 22) indicate time of fermentation. Beer samples sharing the same upper letter (A, B, C, D, E, F, G) and roman numeral (I, II) belong to the same sub-cluster and cluster respectively.

## 4. Discussion

Analysis of sorghum grains used for beer production showed that they did not contain enough impurities and were significantly dried. The impurity content of raw material was lower than 5%, which is the recommended value of the acceptability of grains [20]. The water content of grains that was lower than 13% means that the sorghum grains were well-dried and kept before selling. Grains used for processing the indigenous alcoholic beverages were above the recommended values (25–29 g/1000 grains) indicated by the Food and Agriculture Organization (FAO) [21]. However, they were in the same weight range with the sorghum grains commonly used for the production of indigenous beers in northern Cameroon [22]. A significant gap was observed between the germination energy of sorghum grains obtained with 4 and 8 mL of water, which confirms the key role played by water during the malting step. With all these physicochemical and technological data,

we postulated that sorghum grains used for brewing the indigenous beers were suitable for malting and the quality of wort that could be obtained after brewing may be considerable. Indeed, the wort yields obtained in this study were higher than those obtained of around 56% during the production of sorghum-based turbid beers and indigenous beers based on sorghum and banana brewed in West African countries [23]. However, the wort yields of sorghum beers produced in northern Cameroon were close to those reported of about 60 to 64% for the production of plantain sorghum-based beer and alcoholic beverage based pure plantain in the Ivory Coast [24]. Physicochemical changes during fermentation for both sorghum turbid beers show that it depends on the raw material and production process used, but also on the use or not of a starter. Significant decrease in pH observed after a day of fermentation might be explained by the presence of basic amino acid content in the wort. The level of these nitrogen compounds and sugar favor yeast metabolism and accounts on the type of fermentation occurring. According to Akin [25], a large content of wort in basic amino acids such as arginine and lysine and their consumption by yeast during metabolism causes a release of a large number of protons equivalent to the positive charge of the basic amino acids. Furthermore, the production of organic acids and carbon dioxide by the consumption of sugars during early lactic fermentation and late alcoholic fermentation, respectively, contribute in the pH reduction from the first hours of fermentation [26]. This was confirmed by a significant rise of acetic and malic acid content after the first day of fermentation. Nevertheless, the non-significant increase in pH after the first day was due to some physicochemical modifications that occurred during fermentation. This result agrees with the report by Maoura et al. [27], who showed that pH did not increase significantly during fermentation of bili bili, a sorghum turbid beer home-brewed in Chad. Indeed, the production of alcohol from sugar during fermentation results in a decrease in organic acids dissociation, which induces a lower proton release and therefore a slight increase in pH [25]. The relatively high content of residual sugars after fermentation can be explained by a limited consumption of sugars by yeasts [28] and the high initial total soluble solids content of the worts. It contributes to increase of osmotic pressure and reduce of the performance of yeast [29]. Changes in alcohol, total solids and sugar contents clearly show and confirm the fact that production of alcohol during fermentation goes with consumption of the reducing sugars contained in the wort by bacteria and yeasts. Changes in temperature during fermentation is associated with the metabolism of the fermentative microorganisms which use sugars and grow up following three particular steps: acceleration, deceleration and stabilization [30]. Higher apparent conductivity of the red beer samples compared to the white beer is justified by minerals contents of the raw material used for the production of each indigenous sorghum-based alcoholic beverage. Grains of the red variety used for the preparation of the red turbid beer are richer in calcium, iron, magnesium, and phosphorus than grains of the white variety used for the white home-brewed beer [31]. This is due to the high content of the antinutritional factors such as phytate and tannins in the white sorghum grains which decrease the bioavailability of minerals [32].

The principal component analysis (PCA) performed on some physicochemical variables of sorghum turbid beers produced in the Northern regions of Cameroon clearly shows that there is a positive correlation between sugars, total soluble solids (TSS), density, and pH on one hand, and between organic acids and alcohol contents on the other hand, and all these variables were loaded on the principal component F1. Indeed, factor analysis indicates that consumption of sugars during fermentation is accompanied by a drop in density, total soluble solids (TSS), and especially pH, justifying the acidic character of the indigenous turbid beers at the end of fermentation. The increase in malic and acetic acid contents during fermentation is accompanied by a rise in alcohol production. These findings disagree with the report of Panda et al. [33] on home-brewed beers based on barley and anthocyanin-rich sweet potato produced in India. They showed that pH and total soluble solids (TSS) loaded on the principal component PC1 were negatively correlated as well as ethanol and organic acids, both loaded on the principal component PC2. Density and sugar content were loaded on the third main component PC3. This difference could

be explained by the nature of the raw material (sorghum or barley) and the quality of the starter (traditional or industrial) used for the production of alcoholic beverages. The difference in changes of the physicochemical parameters was confirmed by loading of the measured variables on principal component PC1. For example, alcohol and organic acid contents were positively loaded on principal component PC1 for the white sorghum turbid beer samples, whereas they contributed negatively to PC1 for the red sorghum turbid beer samples. The same observations were seen for the remaining physicochemical variables. A report on the two main clusters shown by hierarchical analysis (Figure 10) confirmed that the indigenous turbid sorghum beer samples before and after the fermentation process were deeply different and belonged to two distinct groups. However, the red and white sorghum beer samples were respectively grouped into two and three main sub-clusters. Indeed, red turbid beers fermented for a short time (one to two days) were clearly distinguished from those fermented for a long time (three to 10 days) while the white turbid beers fermented for one to four days, five to six days, and seven to 10 days were separately grouped. This suggests that the samples belonging to each sub-cluster were closely related. This difference can be explained by the nature and time of fermentation carried out on each of the sorghum turbid beers produced in the northern region of Cameroon. Alcoholic fermentation of the indigenous red sorghum beer occurred faster (addition of a starter) and lasted longer than 24 h before the beginning of consumption, while alcoholic fermentation of the white sorghum beer was usually preceded by lactic fermentation, which is slow (absence of a starter) because it lasts for about 48 to 72 h before consumption of the beverage. In artisanal fermented products, biological risks such as pathogenic microorganisms as well as toxic molecules from fungal origin are possible as mycotoxins can be found [12,13,34]. Many studies have previously reported the presence of mycotoxins in artisanal beers traditionally brewed in South Africa [35], maize-based turbid beers produced in Cameroon [36] and Malawi [37], and opaque sorghum and millet beers from Kenya [38]. The presence of mycotoxins in traditional fermented cereal-based beverages is a public health concern considering the adverse effects of these toxins. However, the mycotoxin content of cereals (maize, millet and sorghum) is generally higher than those of traditional fermented cereal-based beverages considered as marginal. This reduction in mycotoxins, which can reach up to 95% in some traditional brewed beers such as *pito*, depends on the kind of processing techniques employed [39]. It has been suggested that fermentation and other processing techniques such as overnight soaking and milling of grains, boiling, and sieving to get a drinkable filtrate, are able to make beverages relatively safe for consumption in terms of mycotoxin content. Furthermore, it should be noted that lactic acid bacteria (LAB), which is predominant in some wild cultures used as starters for non-alcoholic fermentation, can degrade some mycotoxins such as aflatoxin by their ability to bind, degrade, or inhibit aflatoxin biosynthesis in food matrices [40].

## 5. Conclusions

The level of physicochemical changes during fermentation of the indigenous alcoholic sorghum-based beverages brewed in the northern regions of Cameroon and the correlation between some physicochemical parameters and various fermented alcoholic sorghum turbid beverages were highlighted in this work. It was shown that the physicochemical parameters significantly changed just after the first or second day of fermentation; after that, there were no significant physicochemical changes. The correlation between the physicochemical variables was demonstrated using the factor analysis loading plots. Hierarchical clustering analysis grouped sorghum turbid beers based on their physicochemical profiles and time of fermentation into two principal clusters. The results reported could serve as a starting point to well understand the fermentation process during the production of indigenous alcoholic sorghum-based beverages and to improve the efficiency of the process. However, further experiments based on hazard analysis are needed to study the microbial changes, to determine critical contamination steps during processing that could

significantly affect the hygiene quality and physicochemical parameters of indigenous sorghum turbid beers.

**Author Contributions:** Conceptualization, J.R.B. and F.-X.E.; Investigation, Methodology, Formal analysis, Writing—Original Draft, J.R.B.; Supervision and Writing—Review & Editing, F.-X.E. All authors have read and agreed to the published version of the manuscript.

**Funding:** This research received no external funding.

**Conflicts of Interest:** The authors declare no conflict of interest.

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
