# Peer review of "Physicochemical Changes Occurring during Long-Time Fermentation of the Indigenous Alcoholic Sorghum-Based Beverages Brewed in Northern Cameroon"

_beverages, doi:10.3390/beverages7020039_

Round 1

Reviewer 1 Report

See attached document!

Author Response

The authors thank the reviewer for his/her noteworthy work, inputs and comments.  we hope that we could clarify the open points and we strongly believe that the quality of the paper could be further increased. The attached document provides a point by point response for each reviewers’ comments.

Reviewer 2 Report

The authors describe an interesting and novel research into the changes during artisanal fermentation of Cameroon beer.

The following changes are required:

Line 12: Is it really the main consumed drink? Consider drinking water, tea, coffee etc

Line 69: Is “leaven” a word?

Line 102: add references for previous studies

Section 2.4.: add more details, weights, voluminal etc. The processing method must be replicable.

Line 135: use decimal point not comma

Materials  and methods throughout: please give city/country for all suppliers/manufacturers

Throughout: check author guideline for intext references (should be numbered consecutively, not author/date)

Line 320: Mycotoxins as potential impurities in African beers could be discussed. Check Okaru et al. https://www.sciencedirect.com/science/article/abs/pii/S0956713517301913

Author Response

The authors thank the reviewer for his/her noteworthy work, inputs and comments. We hope that we could clarify the open points and we strongly believe that the quality of the paper could be further increased. The attached document provides a point by point response for each reviewers’ comments.

Round 2

Reviewer 1 Report

See attached document!

Author Response

The authors are very thankful for the new inputs and comments of the reviewer. They are open for any further suggestions.
